# VideoGPT+ 🤖: Integrating Image and Video Encoders for Enhanced Video Understanding

## Abstract

Building on the advances of language models, Large Multimodal Models (LMMs) have contributed significant improvements in video understanding. While the current video LMMs utilize advanced Large Language Models (LLMs), they rely on either image or video encoders to process visual inputs, each of which has its own limitations. Image encoders excel at capturing rich spatial details from frame sequences but lack explicit temporal context, which can be important in videos with intricate action sequences. On the other hand, video encoders provide temporal context but are often limited by computational constraints that lead to processing only sparse frames at lower resolutions, resulting in reduced contextual and spatial understanding. To this end, we introduce `VideoGPT+`, which combines the complementary benefits of the image encoder (for detailed spatial understanding) and the video encoder (for global temporal context modeling). The model processes videos by dividing them into smaller segments and applies an adaptive pooling strategy on features extracted by both image and video encoders. Our architecture showcases improved performance across multiple video benchmarks, including VCGBench, MVBench and Zero-shot question-answering. Further, we develop 112K video-instruction set using a novel semi-automatic annotation pipeline which further improves the model performance. Additionally, to comprehensively evaluate video LMMs, we present `VCGBench-Diverse`, covering 18 broad video categories such as lifestyle, sports, science, gaming, and surveillance videos. This benchmark with 4,354 question-answer pairs evaluates the generalization of existing LMMs on dense video captioning, spatial and temporal understanding, and complex reasoning, ensuring comprehensive assessment across diverse video types and dynamics. Our code, dataset, and pre-trained models will be publicly released.

## 1 Introduction

Existing methods for video understanding often rely solely on either image encoders or video encoders (Maaz et al., 2024; Jin et al., 2024; Liu et al., 2024c). Most works focus on image encoders, which encode multiple frames and either fuse the information or concatenate the embeddings before passing them to the LLM. When fusing the information, spatial or temporal pooling is typically used (Maaz et al., 2024). Spatial pooling has shown minimal effectiveness in capturing video information, whereas temporal pooling retains some spatial information but lacks explicit temporal context. On the other hand, concatenating embeddings without pooling (Jin et al., 2024; Liu et al., 2024c; Zhang et al., 2024b) can rapidly increase computational complexity due to the extended context length required by the LLM, limiting the number of frames that can be processed. While this approach provides better spatial representation, the overall context is still limited to few frames. The limited context results in a poor understanding of the video, especially if a uniform sampling strategy is employed, as it only captures small segments of the video, missing important temporal dynamics.

In order to address these challenges, we propose `VideoGPT+` which effectively combines the merits of both image and video encoders (see Fig. 2). By leveraging an image encoder for rich spatial details and a video encoder for global temporal context, our model achieves improved video understanding. To model finegrained temporal dynamics in `VideoGPT+`, we use a segment-wise sampling strategy. Unlike uniform sampling used in existing video LMMs (Maaz et al., 2024), which may miss important temporal dynamics, our approach divides the video into smaller segments and

applies segment-wise sampling. This ensures that the model captures representative information from different segments of the video, enabling a more comprehensive understanding.

To facilitate the integration of image and video features, `VideoGPT+` introduces a visual adapter module that combines their complimentary benefits. This module performs projection and pooling operations, mapping both image and video features to a common space while reducing computational complexity. By aligning the features in this manner, the model can effectively utilize the combined spatial and temporal information for improved video understanding.

We demonstrate the effectiveness of `VideoGPT+` across five standard video-conversation benchmarks, including VCGBench (Maaz et al., 2024), MVBench (Li et al., 2024), and Zero-shot question-answering (Maaz et al., 2024), where it outperforms previous SoTA approaches (see Fig. 1). Further, we develop `VCG+112K` using a novel semi-automatic annotation pipeline (see Fig. 3), which provides dense video captions along with spatial understanding and reasoning-based question-answer (QA) pairs, further enhancing the model's performance. We also propose `VCGBench-Diverse`, extending VCGBench (Maaz et al., 2024) by including videos from 18 different domains to extensively evaluate the video-based conversation models in diverse domains (see Fig. 4).

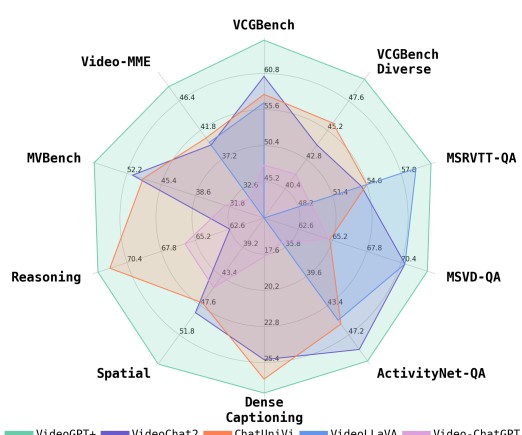

Figure 1: **Performance comparison of VideoGPT+** with various SoTA models on multiple video benchmarks. `VideoGPT+` permors better compared to various models (Li et al., 2023c; Jin et al., 2024; Lin et al., 2023; Maaz et al., 2024) on video conversation benchmarks: VCGBench (Maaz et al., 2024), Video-MME (Fu et al., 2024), MVBench (Li et al., 2023c) and Zero-shot video QA: MSVD-QA, MSRVTT-QA, ActivityNet-QA. We also evaluate on `VCGBench-Diverse` that covers 18 broad video categories (across dense captioning, spatial understanding, and reasoning).

Our work has three main contributions:

- We present `VideoGPT+`, the first video-conversation model that benefits from a dual-encoding scheme based on both image and video features. These complimentary sets of features offer rich spatiotemporal details for improved video understanding (Sec. 3).
- Addressing the limitations of existing VideoInstruct100K dataset (Maaz et al., 2024), we develop `VCG+112K` with a novel semi-automatic annotation pipeline, offering dense video captions along with spatial understanding and reasoning-based QA pairs, improving model performance (Sec. 4).
- Recognizing the lack of diverse benchmarks for video-conversation task, we propose `VCGBench-Diverse`, which provides 4,354 human annotated QA pairs across 18 video categories to extensively evaluate the performance of a video-conversation model (Sec. 5).

## 2 RELATED WORKS

Building on advances in language models, LLMs offer a flexible interface for various multimodal applications. Early efforts in image-based conversation models such as BLIP-2 (Li et al., 2023b), MiniGPT-4 (Zhu et al., 2024) and LLaVA (Liu et al., 2023c;b) project image features into the language space through a learnable module and perform instruction tuning for visual conversations capabilities. Other efforts extend these models to visual grounding tasks (Peng et al., 2023; Rasheed et al., 2024; You et al., 2023), exploring the potential of LLMs in complex vision tasks.

**Video Conversation Models:** Initial works like Video-ChatGPT (Maaz et al., 2024) and Video-LLaMA (Zhang et al., 2023) extend image-based models to the video domain by introducing components to encode temporal features, where frame-level visual features are fed to the LLM. However, this is computationally expensive and quickly fills its context window. To address this issue, Video-ChatGPT (Maaz et al., 2024) employs spatial and temporal pooling. LLaMA-Vid (Li et al., 2023d) proposes representing a single image with two tokens, context and content. IG-VLM (Kim et al., 2024) treats a video as a grid of images, while LITA (Huang et al., 2024b) employs slow-fast

token pooling to reduce the number of visual features. Chat-UniVi (Jin et al., 2024) uses clustering in both spatial and temporal dimensions to merge tokens, and VideoChat (Li et al., 2023c) uses Q-Former (Li et al., 2023b) to learn a fixed number of queries by cross-attending to the visual features. MobileVLM (Chu et al., 2023; 2024) utilize a lightweight CNN to reduce the spatial dimensions. Other notable methods include (Liu et al., 2024b; Lin et al., 2023; Munasinghe et al., 2023; Song et al., 2024; Huang et al., 2024a).

Alternatively, methods such as VideoChat2 (Li et al., 2024) use pretrained video encoders. Although video encoders provide temporal context, they are limited by computational constraints, operating with limited frames at lower resolutions, restricting temporal context and spatial understanding. Our `VideoGPT+` model addresses these issues by using segment-wise sampling and effectively combining image and video encoders to capture rich spatial and temporal details (see Fig. 2).

**Video Instruction Tuning Datasets:** VideoChat (Li et al., 2023c) builds a video-instruction tuning dataset consisting of 7K instructions using videos from WebVid-10M (Bain et al., 2021). Video-ChatGPT (Maaz et al., 2024) introduces a semi-automatic annotation pipeline to generate VideoInstruct100K using videos from ActivityNet (Fabian Caba Heilbron & Niebles, 2015). VideoChat2 (Li et al., 2024) combines multiple existing image and video datasets to develop a 1.9M joint image-video instruction tuning dataset. In our experiments, we use VideoInstruct100K and a subset of the dataset from VideoChat2. Additionally, addressing the limitations of the VideoInstruct100K dataset (Maaz et al., 2024), we develop `VCG+112K` through a novel semi-automatic annotation pipeline, which provides dense video captions along with 112K QA pairs targeting reasoning, spatial and temporal understanding, which further improves model's understanding of video content (see Fig. 3).

**Video Conversation Benchmarks:** Video-ChatGPT (Maaz et al., 2024) introduces VCGBench and zero-shot QA benchmarks, where VCGBench includes 500 videos with 3000 QA pairs, evaluated using GPT-3.5 across various metrics. Despite its comprehensive evaluation, it only contains videos from the ActivityNet dataset. The Zero-shot evaluation covers MSVD-QA (Xu et al., 2017), MSR-VTT-QA (Xu et al., 2017), TGIF-QA (Jang et al., 2019), and ActivityNet-QA (Fabian Caba Heilbron & Niebles, 2015). MVBench (Li et al., 2024) consists of 4K QA pairs evaluating 20 temporal tasks, though it mostly includes short videos averaging 5-40 seconds. Another recent benchmark, Video-MME (Fu et al., 2024), addresses the issue of diversity by incorporating a wide range of videos. However, both MVBench and Video-MME are limited to MCQs, which, while straightforward for evaluation, restrict the range of questions that can be asked and reduce the depth of understanding the model can demonstrate. By confining to predefined choices, MCQs introduce bias and fail to capture the model's true understanding. Considering the limitation of existing benchmarks, which often lack focus on generalization and diversity, we propose `VCGBench-Diverse`, featuring 4,354 QA pairs from 877 videos across 18 domains, evaluated using open-ended questions (see Fig. 4).

## 3 METHOD

For effective video understanding, combining detailed spatial information with explicit temporal context is crucial. To achieve this, we propose `VideoGPT+`, which features a dual encoder design that leverages the complementary strengths of an image encoder and a video encoder.

**Overall Architecture**: The overall architecture consists of (i) segment-wise sampling, (ii) dual visual encoder, (iii) vision-language adapters that project vision features to the language domain and (iv) a large language model. Frames selected through a segment-wise sampling strategy are encoded through a dual encoder consisting of an image and a video encoder. Both sets of features are projected to language space using vision-language (V-L) adapters, and the resulting tokens are pooled through adaptive token pooling and concatenated before being fed to the LLM (see Fig. 2).

**Segment-wise Sampling:** To extract fine-grained temporal cues, we use a segment-wise frame sampling strategy. Given an input video $\mathbf{V} \in \mathbb{R}^{T \times H \times W \times C}$, we divide it into $K$ segments, where each segment consists of $n = \frac{T}{K}$ frames. Thus, the video can be represented as $\mathbf{V} = [\mathbf{V}_k]_{k=1}^{K}$. Each segment $\mathbf{V}_k \in \mathbb{R}^{n \times H \times W \times C}$ can be described as a sequence of frames, $\mathbf{X}_i$, where $\mathbf{V}_k = [\mathbf{X}_{i,j}]_{j=1}^{n}$. The video segments are downsampled to a lower resolution of $n \times h \times w \times c$ for video encoding.

Compared to a uniform sampling, segment-wise sampling better aligns with our dual encoder design. Video encoders often face computational constraints, limiting them to processing only sparse frames. Uniform sampling increases the self-attention computation complexity as it requires attending to

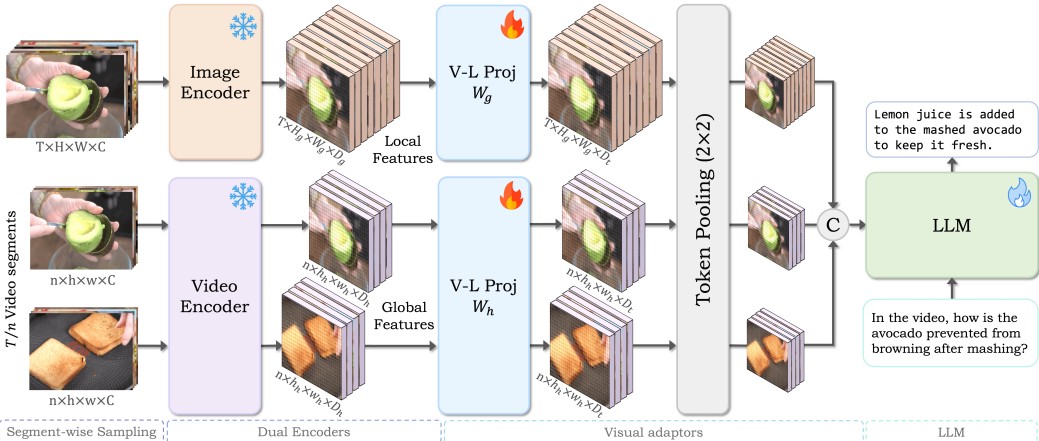

Figure 2: **Overview of VideoGPT+.** `VideoGPT+` is a large multimodal model for video understanding. It uses a dual-encoder design that combines the complementary strengths of an image encoder and a video encoder. The image encoder captures detailed spatial features, while the video encoder captures temporal dynamics across multiple frames. To retain fine-grained temporal details while ensuring efficiency, we use segment-wise frame sampling instead of random sparse sampling. Both sets of features are then projected into a unified space through Vision-Language (V-L) projection layers and the resulting tokens are pooled and concatenated before being processed by a Large Language Model to generate comprehensive responses to video-based questions. Symbols ❄ indicates frozen components, 🔥 indicates trainable components, and the 🔥 indicates LoRA-training.

features of all frames. Additionally, video encoders are typically trained with sparse frames, and providing more frames can hinder their ability to accurately capture temporal information. In contrast, the segment-wise sampling strategy divides the video into smaller, manageable segments, enabling the video encoder to efficiently capture rich temporal cues within each segment.

**Dual Vision Encoder:** Our design leverages the complementary strengths of an image encoder that captures detailed spatial features and a video encoder that provides explicit temporal context. The image encoder $g$, processes $T$ frames, $g(\mathbf{X}) \in \mathbb{R}^{T \times H_g \times W_g \times D_g}$, producing local features that provide frame-level context. Meanwhile, the video encoder $h$, operates on low-resolution video segments $\mathbf{V}_k$, yielding global features that provide segment-wise context, $h(\mathbf{V}_k) \in \mathbb{R}^{n \times h_h \times w_h \times D_h}$.

The primary goal of `VideoGPT+` is to leverage the capabilities of a pre-trained LLM alongside visual modalities from both a pre-trained image encoder and a pre-trained video encoder. Specifically, we utilize the pre-trained CLIP model, ViT-L/14 ($336 \times 336$) (Radford et al., 2021) as the image encoder, and InternVideo-v2 ($224 \times 224$) (Wang et al., 2024) as the video encoder. These models are selected for their robust performance and their ability to complement each other in capturing both spatial and temporal information. Both encoders are pre-trained on large-scale datasets in a multimodal setting using contrastive loss, facilitating their integration within our architecture.

**Visual Adapter:** The output embeddings from the second last layer of both image and video encoders are passed through separate V-L projection layers, $W_g$ and $W_h$, respectively. These Multi-Layer perceptrons (MLPs) project the visual features into the language space. The projection layers are trainable, while the visual encoders remain frozen, preserving the rich, pre-trained representations. The projected embeddings are reshaped back into their grid forms and subjected to a $2 \times 2$ adaptive token pooling, which operates on the spatial dimensions of the local and global features. This pooling reduces the token length by a factor of $4$, thereby allowing to fit in larger visual context within the same LLM context window. The pooled embeddings from the local features form $\mathbf{E}^{img} \in \mathbb{R}^{T \times h_g \times w_g \times D_t}$, while the pooled embeddings from the global features of each segment form $\mathbf{E}^{vid} \in \mathbb{R}^{n \times h_h \times w_h \times D_t}$.

**Large Language Model:** We obtain the final representation by concatenating the embeddings $\mathbf{E}^{img}$ with $K$ segment-wise embeddings $\mathbf{E}^{vid}$, such that we have detailed spatial representation across all segments followed by their global temporal context. We then concatenate the text embeddings $\mathbf{E}^{text} \in \mathbb{R}^{L \times D_t}$ of the user text query with the visual embeddings,

$$\mathbf{E} = [\mathbf{E}^{img}, \mathbf{E}^{vid}_1, \dots, \mathbf{E}^{vid}_K, \mathbf{E}^{text}]. \tag{1}$$

This integration ensures that the LLM receives a sequence of embeddings that include detailed spatial features from the image encoder and comprehensive temporal context from the video encoder,

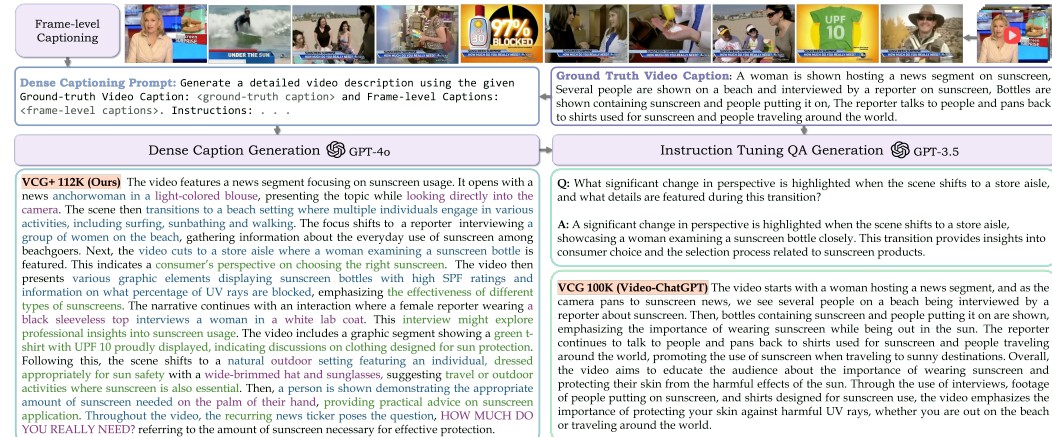

Figure 3: **Illustration of the semi-automatic annotation process in VCG+ 112K**. The figure shows how we use ground-truth video captions and frame-level descriptions to generate a detailed video description. GPT-4 is used to remove irrelevant and conflicting noisy information in the frame-level descriptions to produce a high-quality video description. The semi-automatic annotation process integrates spatial, temporal and event, and reasoning details into the brief information we start with. This dense video description is then used to generate instruction-tuning QA pairs using GPT-3.5. We provide detailed prompts used in both stages in Appendix D (see Figs. 8 and 9). We also compare the video description in the VideoInstruct100K (Maaz et al., 2024) dataset to show the improvement in quality achieved by our new annotation pipeline.

allowing for robust video understanding. The LLM is fine-tuned using LoRA (Hu et al., 2021) in an auto-regressive manner with a next-token prediction loss. Refer to Fig. 2 for detailed illustration.

## 4 DATASET

Video-ChatGPT (Maaz et al., 2024) introduces the VideoInstruct100K dataset, which employs a semi-automatic annotation pipeline to generate 75K instruction-tuning QA pairs. To address the limitations of this annotation process, we present `VCG+ 112K` dataset developed through an improved annotation pipeline. Our approach improves the accuracy and quality of instruction tuning pairs by improving keyframe extraction, leveraging SoTA large multimodal models (LMMs) for detailed descriptions, and refining the instruction generation strategy.

**Keyframe Extraction**: VideoInstruct100K uses a fixed number of video keyframes, regardless of video length or dynamics, to generate frame-level dense captions. This often results in both insufficient and redundant information. We address this by first extracting scenes from videos (Castellano, 2022), and then selecting one keyframe/scene. Consequently, we obtain detailed information for videos with rich content and reduce redundancy for videos with less content. It provides better visual context by extracting more stable keyframes, thus offering a more accurate video representation.

**Frame-Level Descriptions**: After extracting keyframes, we use a SoTA image LMM, LLaVA-v1.6 (Liu et al., 2024a), to generate dense descriptions for each keyframe. These descriptions encompass comprehensive visual details, including spatial attributes, scene context, and object characteristics, which are often absent in concise ground truth captions. While ground truth captions are precise, they lack the granularity to capture intricate visual and spatial information. To address this, we augment them captions with detailed but noisy information from the frame-level descriptions, thus enhancing the quality and accuracy of the subsequent video descriptions.

**Detailed Video Descriptions**: VideoInstruct100K (Maaz et al., 2024) prompts GPT-3.5 directly with frame-level descriptions and concise ground truth captions to generate QA pairs, imposing a significant cognitive load on the model to verify frame-level descriptions with the ground truth. We improve this process by first creating a coherent and detailed video description. We prompt GPT-4 to integrate the detailed frame-level descriptions with the ground truth captions by comparing information and removing any inconsistencies. The resulting detailed descriptions include a timeline of events, actions, object attributes, and scene settings, providing a thorough representation of the video content. This structured input simplifies the task for LLM, thereby enhancing the generated QA pairs quality.

| Category | Description | # | Domains |
|---|---|---|---|
| **5 Video Capturing Methods** | | | |
| Stable Settings | Videos shot in stable, predictable environments with minimal camera movement. | 1200 | Cooking, How-to, Education |
| Dynamic Settings | Videos with significant camera movement requiring adaptation to rapid context shifts. | 448 | Sports, Traffic, Travel |
| Fixed Cameras | Videos recorded from stationary cameras, providing consistent viewpoints for monitoring purposes. | 124 | Surveillance, Automobile |
| Professional Quality | Professionally produced videos with high visual and audio quality, and controlled lighting. | 1608 | News, Film |
| Variable Quality | Informal videos with varying quality, often using handheld devices, captured in spontaneous settings. | 124 | Lifestyle, Pets |
| **6 Reasoning Complexities** | | | |
| Sequential Understanding | Requires comprehension and following of a series of steps or actions in order. | 828 | Cooking, How-to, Education |
| Predictive Reasoning | Involves understanding and predicting outcomes of dynamic, intricate action sequences. | 180 | Sports, Gaming |
| World Knowledge | Demands integration of broader contextual information and world knowledge to interpret video content. | 848 | Science, News |
| Causal Reasoning | Focuses on understanding cause-and-effect relationships within the video. | 340 | Surveillance, Activism |
| Emotional Reasoning | Involves interpreting stories, character motivations, and emotional subtexts. | 1080 | Entertainment, Film, Comedy |
| Analytical Reasoning | Requires critical analysis and interpretation of complex information or situations. | 228 | Traffic, Automobile |

Figure 4: **Illustration of VCGBench-Diverse video conversational benchmark**. `VCGBench-Diverse` comprehensive benchmark is designed to evaluate video LMMs across 18 broad video categories. With 4,354 QA pairs, `VCGBench-Diverse` tests generalization on dense video captioning, spatial and temporal understanding, and complex reasoning. It covers five video-capturing methods, ensuring diversity and robust generalization and six reasoning complexities, assessing various analytical and comprehension skills.

**Improved Instruction Tuning Data**: Using the ground truth captions and detailed video descriptions, we generate two types of high-quality QA pairs using GPT-3.5: descriptive and concise. For **descriptive** instruction pairs, we focus on three categories: (i) *dense captioning*, which provides descriptions of the video covering the entire sequence of events and visual details; (ii) *detailed temporal information*, which addresses the sequence of events and their dependency to learn temporal relationships; and (iii) *generic question answering*, which involves in-depth questions about different actions, their consequences, and other detailed aspects of the video. For **concise** instruction pairs, we target (i) *spatial reasoning*, focusing on understanding and describing spatial details such as scene settings, number of objects, attire, and locations; (ii) *reasoning* of events, covering the causal relationships between events; and (iii) *short temporal questions*, addressing specific moments or sequences, such as what happened at the beginning or end.

# 5 PROPOSED BENCHMARK

Recognizing the limited diversity in existing video conversation benchmarks, we introduce `VCGBench-Diverse` to comprehensively evaluate generalization ability of video LMMs. While VCG-Bench (Maaz et al., 2024) provides an extensive evaluation protocol, it is limited to videos from the ActivityNet200 (Fabian Caba Heilbron & Niebles, 2015) dataset. Our benchmark comprises a total of 877 videos, 18 broad video categories and 4,354 QA pairs, ensuring a robust evaluation framework. The detailed breakdown of `VCGBench-Diverse` is illustrated in Fig. 4, showcasing the distribution of videos across content domains, video capturing methods, and reasoning complexities.

We collect videos from *18 distinct domains*, including lifestyle, how-to, science and technology, news, travel, entertainment, film, sports, comedy, activism, gaming, education, surveillance, pets, cooking, music, automobile, and traffic These categories encompass a broad spectrum of real-world scenarios, ensuring that models are evaluated on a diverse set of challenges. In addition to content diversity, `VCGBench-Diverse` includes a variety of *video capture methods*, which ensures a comprehensive assessment of robustness to different filming techniques, camera movements, quality levels and lighting. The benchmark covers *five* video capture methods including static and controlled settings, dynamic and unpredictable settings, fixed camera perspectives, professional and high-quality videos, and uncontrolled and variable quality. Further, the benchmark evaluates models across *six reasoning complexities*, including sequential understanding, complex action and predictive reasoning, contextual and world knowledge reasoning, causal reasoning, narrative and emotional reasoning, and analytical and critical reasoning, which is crucial for understanding diverse video content.

The videos in `VCGBench-Diverse` are sourced from HDVILA (Xue et al., 2022), MPII (Andriluka et al., 2014), YouCook2 (Zhou et al., 2018), UCF Crime (Sultani et al., 2018), and STUD Traffic (Xu et al., 2021). The video durations range from 29 sec to 471 sec, with an average of 217 sec. Human annotators are tasked with writing detailed descriptions based on their understanding of both audio and visual elements of the videos. This comprehensive annotation process involves a set of annotators

who are provided with an initial set of ten videos each. These annotations undergo a meta-review stage where feedback is provided, and necessary corrections are made to meet the required standards. Following this, annotators receive additional batches, with random samples being selected for quality checks by the meta-reviewer. The final human annotations are utilized to generate QA pairs using GPT-3.5, based on prompts detailed in Fig. 10.

Following VCG-Bench (Maaz et al., 2024), the evaluation is computed over five different aspects: (i) correctness of information (ii) detail orientation (iii) contextual understanding (iv) temporal understanding and (v) consistency. Additionally, `VCGBench-Diverse` provides a breakdown of performance across three key aspects: (i) dense video captioning, which assesses the ability to generate detailed and accurate descriptions of the video content, (ii) spatial understanding, which evaluates the capability to understand and describe the spatial relationships and settings within the video, and (iii) reasoning, which tests the adeptness in inferring and explaining causal relationships and actions within the video.

## 6 EXPERIMENTS

We perform quantitative evaluation of `VideoGPT+` on *five standard benchmarks*: i) VCG-Bench (Maaz et al., 2024), ii) `VCGBench-Diverse`, iii) MVBench (Li et al., 2024), iv) Video-MME (Fu et al., 2024) and v) Zero-shot QA.

**Implementation Details:** We use CLIP-L/14 (Radford et al., 2021) as our image encoder, InternVideo-v2 (Wang et al., 2024) stage-2 1B model as our video encoder in conjunction with Phi-3-Mini-3.8B (Abdin et al., 2024) based LLM with 4K context window in our experiments. The image encoder operates at $336 \times 336$, while the video encoder operates at $224 \times 224$ resolution. Our training consists of two pretraining stages and one instruction-tuning stage. In the pretraining stage, we train with only the image encoder and only the video encoder on the CC-595K dataset (Liu et al., 2023a), with only the visual adapters being learned while the rest of the model is kept frozen. During the instruction-tuning stage, we use LoRA (Hu et al., 2022) with $r = 64$ for LLM, while visual adapters are fully trained and vision encoders are kept frozen. The LR is set to $1e^{-3}$ during pretraining and $2e^{-4}$ during instruction tuning.

For experiments on VCGBench, `VCGBench-Diverse` and Zero-shot QA, we sample 16 frames from videos, while for MVBench which consists of relatively shorter videos, we sample 8 frames. We keep the same sampling strategy during inference. For VCGBench and `VCGBench-Diverse`, the model is trained on VideoInstruct100K (Maaz et al., 2024), `VCG+112K`, conversation and caption data from VideoChat (Li et al., 2023c) and VQA dataset from WebVid (Bain et al., 2021), that combines to approximately 260K single turn conversations. For MVBench, the model is trained on Kinetics-710 (Kay et al., 2017), Something-Something-v2 (Goyal et al., 2017), conversations from VideoChat (Li et al., 2023c), CLEVRER (Yi et al., 2019), VQA dataset from WebVid (Bain et al., 2021) and NExT-QA (Xiao et al., 2021) datasets, which combines to approximately 330K single turn conversations. We run all trainings for one epoch. Following previous approaches (Maaz et al., 2024; Jin et al., 2024; Liu et al., 2024c), we employ GPT-3.5-Turbo-0613 for VCGBench and Zero-shot QA evaluation. However, for our proposed `VCGBench-Diverse`, we employ the latest GPT-3.5-Turbo-0125 for evaluation.

**VCGBench:** The benchmark consists of around 3000 QA pairs generated from 500 human-annotated videos. It evaluates responses based on five aspects: i) CI (Correctness of Information) - accuracy of the response with video content, ii) DO (Detail Orientation) - depth of the response, iii) CU (Contextual Understanding) - alignment with video context, iv) TU (Temporal Understanding) - accuracy in identifying temporal sequences, and v) CO (Consistency) - response consistency to similar questions. Table 1

| Method | CI | DO | CU | TU | CO | Avg. |
|---|---|---|---|---|---|---|
| Video-ChatGPT | 2.40 | 2.52 | 2.62 | 1.98 | 2.37 | 2.38 |
| BT-Adapter | 2.68 | 2.69 | 3.27 | 2.34 | 2.46 | 2.69 |
| VTimeLLM | 2.78 | 3.10 | 3.40 | 2.49 | 2.47 | 2.85 |
| Chat-UniVi | 2.89 | 2.91 | 3.46 | **2.89** | 2.81 | 2.99 |
| LLAMA-VID | 2.96 | 3.00 | 3.53 | 2.46 | 2.51 | 2.89 |
| Video-LLaVA | 2.84 | 2.86 | 3.44 | 2.46 | 2.57 | 2.81 |
| VideoChat2 | 3.02 | 2.88 | 3.51 | 2.66 | 2.81 | 2.98 |
| IG-VLM | 3.11 | 2.78 | 3.51 | 2.44 | 3.29 | 3.03 |
| VideoGPT+ | **3.27** | **3.18** | **3.74** | 2.83 | **3.39** | **3.28** |

Table 1: **Performance of VideoGPT+ on VCGBench (Maaz et al., 2024).** All models use 16 frames except Video-ChatGPT and Chat-UniVi which use 100 and 64 frames respectively.

| Method | CI | DO | CU | TU | CO | Avg. | Caption | Spatial | Reasoning |
|---|---|---|---|---|---|---|---|---|---|
| GPT4o-mini-2024-07-18 | 3.06 | 3.05 | 3.43 | 2.67 | 3.47 | 3.14 | 1.82 | 3.16 | 4.19 |
| Gemini-Pro-1.5-Flash-001 | 3.15 | 3.24 | 3.40 | 2.68 | 3.32 | 3.16 | 2.30 | 3.48 | 3.82 |
| Video-ChatGPT (ACL 2024) (Maaz et al., 2024) | 2.07 | 2.42 | 2.46 | 1.39 | 2.06 | 2.08 | 0.89 | 2.25 | 3.60 |
| BT-Adapter (CVPR 2024) (Liu et al., 2024b) | 2.20 | 2.62 | 2.59 | 1.29 | 2.27 | 2.19 | 1.03 | 2.35 | 3.62 |
| VTimeLLM (CVPR 2024) (Huang et al., 2024a) | 2.16 | 2.41 | 2.48 | 1.46 | 2.35 | 2.17 | 1.13 | 2.29 | 3.45 |
| Chat-UniVi (CVPR 2024) (Jin et al., 2024) | 2.29 | 2.56 | 2.66 | 1.56 | 2.36 | 2.29 | 1.33 | 2.36 | 3.59 |
| VideoChat2 (CVPR 2024) (Li et al., 2024) | 2.13 | 2.42 | 2.51 | 1.66 | 2.27 | 2.20 | 1.26 | 2.43 | 3.13 |
| VideoGPT+ (ours) | 2.46 | 2.73 | 2.81 | 1.78 | 2.59 | 2.47 | 1.38 | 2.80 | 3.63 |

Table 2: **Performance of VideoGPT+ on VCGBench-Diverse.** All open-source models use 16 frames except Video-ChatGPT and Chat-UniVi, which use 100 and 64 frames, respectively. The good performance of our model on `VCGBench-Diverse` shows its generalization to diverse scenarios.

compares our model with previous SoTA approaches. `VideoGPT+` achieves an average score of 3.28 surpassing previous best method by a margin of 0.25 (5%).

**VCGBench-Diverse:** We provide a quantitative comparison of `VideoGPT+` against previous SoTA approaches on `VCGBench-Diverse`, which contains 4,354 QA pairs from 877 videos. Following (Maaz et al., 2024), we evaluate the Correctness of Information (CI), Detail Orientation (DO), Contextual Understanding (CU), Temporal Understanding (TU), and Consistency (CO). Additionally, we provide results for dense captioning, spatial understanding, and visual reasoning abilities. The results are presented in Table 2. `VideoGPT+` achieves an average score of 2.47 surpassing all previous methods. Further, we achieves a score of 1.38, 2.80, and 3.63 on dense captioning, spatial understanding, and visual reasoning, respectively. Notably, `VideoGPT+` achieves improvements in spatial and temporal understanding, surpassing previous best models by 0.37 (7.4%) and 0.23 (4.6%), respectively. This is attributed to the dual encoder architecture, where the high-resolution image encoder enhances spatial understanding and the video encoder improves temporal accuracy.

To further validate the alignment of GPT scores with human preferences, we conduct a study involving human annotators. Four annotators given the same GPT scoring guidelines, each reviewed 50 questions from a pool of 200 randomly selected questions. They scored responses from three models: VideoGPT+, VideoChat2, and Chat-UniV. Their respective scores, 2.0, 1.9, and 2.3, closely matched the GPT averages of 2.3, 2.2, and 2.5 for each model. This comparison confirms that GPT scores align well with human preferences, supporting the reliability of our evaluation method.

Table 2 also shows the results of closed-source models in gray for reference. Note that the comparison between open-source and significantly larger, closed-source models is not fair due to the vast differences in scale, parameters, and training data. We compare `VideoGPT+` (3.8B-scale) with similarly scaled open-source models (7B-scale), where our model demonstrates superior performance.

**MVBench:** We evaluate `VideoGPT+` on MVBench (Li et al., 2024), which provides 4,000 QA pairs from 11 video datasets covering a broad spectrum of scenes, ranging from first-person to third-person and from indoor to outdoor environments. The tasks are categorized into 20 fine-grained temporal understanding tasks. The results presented in Table 3 compare `VideoGPT+` with previous methods, indicating an overall improvement of 7.6% compared to the previous best, VideoChat2. Specifically, `VideoGPT+` achieves SoTA results in 14 out of 20 tasks and comes second in 4 out of

| Model | AS | AP | AA | FA | UA | OE | OI | OS | MD | AL | ST | AC | MC | MA | SC | FP | CO | EN | ER | CI | Avg. |
|---|---|---|---|---|---|---|---|---|---|---|---|---|---|---|---|---|---|---|---|---|---|
| Random | 25.0 | 25.0 | 33.3 | 25.0 | 25.0 | 33.3 | 25.0 | 33.3 | 25.0 | 25.0 | 25.0 | 33.3 | 25.0 | 33.3 | 33.3 | 25.0 | 33.3 | 25.0 | 20.0 | 30.9 | 27.3 |
| GPT-4V (OpenAI, 2023) | 55.5 | 63.5 | 72.0 | 46.5 | 73.5 | 18.5 | 59.0 | 29.5 | 12.0 | 40.5 | 83.5 | 39.0 | 12.0 | 22.5 | 45.0 | 47.5 | 52.0 | 31.0 | 59.0 | 11.0 | 43.5 |
| Otter-V (Li et al., 2023a) | 23.0 | 23.0 | 27.5 | 27.0 | 29.5 | 53.0 | 28.0 | 33.0 | 24.5 | 23.5 | 27.5 | 26.0 | 28.5 | 18.0 | 38.5 | 22.0 | 22.0 | 23.5 | 19.0 | 19.5 | 26.8 |
| mPLUG-Owl-V (Ye et al., 2023) | 22.0 | 28.0 | 34.0 | 29.0 | 29.0 | 40.5 | 27.0 | 31.5 | 27.0 | 23.0 | 29.0 | 31.5 | 27.0 | 40.0 | 44.0 | 24.0 | 31.0 | 26.0 | 20.5 | 29.5 | 29.7 |
| Video-ChatGPT (Maaz et al., 2024) | 23.5 | 26.0 | 62.0 | 22.5 | 26.5 | 54.0 | 28.0 | 40.0 | 23.0 | 20.0 | 31.0 | 30.5 | 25.5 | 39.5 | 48.5 | 29.0 | 33.0 | 29.5 | 26.0 | 35.5 | 32.7 |
| VideoLLaMA (Zhang et al., 2023) | 27.5 | 25.5 | 51.0 | 29.0 | 39.0 | 48.0 | 40.5 | 38.0 | 22.5 | 22.5 | 43.0 | 34.0 | 22.5 | 32.5 | 45.5 | 32.5 | 40.0 | 30.0 | 21.0 | 37.0 | 34.1 |
| VideoChat (Li et al., 2023c) | 33.5 | 26.5 | 56.0 | 33.5 | 40.5 | 53.0 | 40.5 | 30.0 | 25.5 | 27.0 | 48.5 | 35.0 | 20.5 | 42.5 | 46.0 | 26.5 | 41.0 | 23.5 | 23.5 | 36.0 | 35.5 |
| VideoChat2 (Li et al., 2024) | 66.0 | 47.5 | 83.5 | 49.5 | 60.0 | 58.0 | 71.5 | 42.5 | 23.0 | 23.0 | 88.5 | 39.0 | 42.0 | 58.5 | 44.0 | 49.0 | 36.5 | 35.0 | 40.5 | 65.5 | 51.1 |
| VideoGPT+ (ours) | 69.0 | 60.0 | 83.0 | 48.5 | 66.5 | 85.5 | 75.5 | 36.0 | 44.0 | 34.0 | 89.5 | 39.5 | 71.0 | 90.5 | 45.0 | 53.0 | 50.0 | 29.5 | 44.0 | 60.0 | 58.7 |

Table 3: **Performance of VideoGPT+ on MVBench.** Following (Li et al., 2024), we evaluate on 20 tasks including AS: Action Sequence, AP: Action Prediction, AA: Action Antonym, FA: Fine-grained Action, UA: Unexpected Action, OE: Object Existence, OI: Object Interaction, OS: Object Shuffle, MD: Moving Direction, AL: Action Localization, ST: Scene Transition, AC: Action Count, MC: Moving Count, MA: Moving Attribute, SC: State Change, FP: Fine-grained Pose, CO: Character Order, EN: Egocentric Navigation, ER: Episodic Reasoning and CI: Counterfactual Inference.

| Model | MSVD-QA | | MSRVTT-QA | | TGIF-QA | | ActivityNet-QA | |
|---|---|---|---|---|---|---|---|---|
| | Accuracy | Score | Accuracy | Score | Accuracy | Score | Accuracy | Score |
| FrozenBiLM (Yang et al., 2022) | 32.2 | – | 16.8 | – | 41.0 | – | 24.7 | – |
| VideoChat (Li et al., 2023c) | 56.3 | 2.8 | 45.0 | 2.5 | 34.4 | 2.3 | 26.5 | 2.2 |
| LLaMA Adapter (Zhang et al., 2024a) | 54.9 | 3.1 | 43.8 | 2.7 | - | - | 34.2 | 2.7 |
| Video-LLaMA (Zhang et al., 2023) | 51.6 | 2.5 | 29.6 | 1.8 | - | - | 12.4 | 1.1 |
| Video-ChatGPT (Maaz et al., 2024) | 64.9 | 3.3 | 49.3 | 2.8 | 51.4 | 3.0 | 35.2 | 2.8 |
| ChatUniVi (Jin et al., 2024) | 65.0 | 3.6 | 54.6 | 3.1 | 60.3 | 3.4 | 45.8 | 3.2 |
| LLaMA-VID (Li et al., 2023d) | 70.0 | _3.7_ | 58.9 | 3.3 | – | – | 47.5 | _3.3_ |
| Video-LLaVA (Lin et al., 2023) | _70.7_ | **3.9** | _59.2_ | _3.5_ | _70.0_ | _4.0_ | 45.3 | _3.3_ |
| VideChat2 (Li et al., 2024) | 70.0 | **3.9** | 54.1 | 3.3 | – | – | _49.1_ | _3.3_ |
| VideoGPT+ (ours) | **72.4** | **3.9** | **60.6** | **3.6** | **74.6** | **4.1** | **50.6** | **3.6** |

Table 4: **Performance of VideoGPT+ on Zero-shot QA.** All the models are evaluated in zero-shot setting where none of the videos were included in the training set. VideoGPT+ achieves good results on all datasets.

20 tasks, obtaining an average score of 58.7% across the 20 tasks. Additionally, VideoGPT+ shows significant improvements in the Action Prediction (+12.5%), Object Existence (OE) (+27.5%), Moving Direction (MD) (+17%), Moving Count (MC) (+29%) and Moving Attributes (MA) (+32%) indicating the rich spatial information and temporal context achieved by our model.

**Video-MME:** We evaluate the performance of our model on Video-MME, a more comprehensive benchmark that assesses video understanding across six domains and 30 subfields through 2700 multiple-choice-qa pairs from 900 videos. It covers a diverse range of video durations, from short, medium, and long videos (11 sec

| Model | Short | Med | Long | Avg |
|---|---|---|---|---|
| Video-LLaVA | 45.3 | 38.0 | 36.2 | 39.9 |
| Qwen-VL-Chat | 46.9 | 38.7 | 37.8 | 41.1 |
| ChatUniVi | 45.7 | 40.3 | 35.8 | 40.6 |
| VideoChat2 | 48.3 | 37.0 | 33.2 | 39.5 |
| VideoGPT+ | **56.4** | **47.2** | **42.5** | **48.7** |

to 1 hour). Our results show that our model achieves superior performance compared to prior SoTA approaches. Specifically, our model performs well across the short, medium, and long video categories, demonstrating strong temporal understanding and effectively capturing long-range dependencies

**Zero-shot Question-Answering:** We provide a quantitative comparison of our method on the zero-shot QA task across four open-ended QA datasets, including MSVD-QA (Xu et al., 2017), MSRVTT-QA (Xu et al., 2017), TGIF-QA (Jang et al., 2019), and ActivityNet-QA (Fabian Caba Heilbron & Niebles, 2015). Results presented in Table 4 show VideoGPT+ achieves superior performance compared to previous methods, indicating its ability to adapt effectively to unseen videos and generate accurate contextually relevant responses in challenging settings.

**Vision Encoder Type:** We ablate our dual visual encoder design in VideoGPT+ . We ablate three settings: using only the image encoder, only the video encoder, and both en-

| Vision Encoder | VCG | VCG-Div | Temporal Score | Spatial Score | GPT4 as Judge | |
|---|---|---|---|---|---|---|
| | | | | | VCG | VCG-Div |
| Image-only | 3.17 | 2.36 | 1.61 | 2.70 | 22 | 28 |
| Video-only | 3.20 | 2.38 | 1.69 | 2.64 | 27 | 30 |
| Dual (ours) | **3.28** | **2.47** | **1.78** | **2.80** | **51** | **42** |

coders. The results shows that our dual encoder design effectively combines both spatial and temporal information and achieves the highest score on both VCGBench and VCGBench-Diverse.

Note that the image encoder operates at a higher resolution of 336×336, while the video encoder operates at 224×224. The image encoder captures better spatial information and fine-grained details, while the video encoder contributes to understanding motion and action sequences. We further verify this on MVBench action categories including action sequence (+3.6%), action antonym (+1.5%), fine-grained action (+1.5%) and unexpected action (+4.0%), where video-only model performs better than the image-only model.

For completeness, we use a best response selection method with GPT4-as-a-judge to evaluate different model designs. Responses from three model variants: image encoder, video encoder and our dual encoder design are presented anonymously to GPT4 alongside the ground truth. The model selects the best response among the three and excludes cases with no clear winner. For VCGBench (VCG), 732 out of 2000 samples were scored, where the dual encoder design was preferred in 51% of cases, compared to 22% for the image encoder and 27% for the video encoder. For VCGBench-Diverse (VCG-Div), 792 out of 4354 samples were scored, with the dual encoder preferred in 42% of cases, compared to 28% for the image encoder and 30% for the video encoder, indicating that our dual encoding design as a clear winner among other uni-encoder alternatives (see Table 6).

**Frame-level and Video-level Feature Fusion:** Though our design uses some known components, their meticulous combination to develop an efficient pipeline for video understand-

ing in MLLMs has not been demonstrated before. We ablate our approach with two alternatives: i) Without segment-wise sampling - resulting in less effective temporal information captured by the video encoder impacting performance; ii) Without adaptive token pooling - which limits the model's ability to utilize the LLM context length effectively, restricting the model to fewer frames. We compare the performance on both VCGBench and `VCGBench-Diverse` benchmarks. The results indicate the effectiveness of our proposed fusion strategy.

| Setting | VCG | VCG-Div |
|---|---|---|
| w/o Segment-wise Sampling | 3.21 | 2.40 |
| w/o Adaptive Pooling | 3.08 | 2.31 |
| Video-GPT+ (ours) | **3.28** | **2.47** |

**Pooling Strategy:** We ablate different pooling strategies for the image and video encoders. The image encoder outputs a $24 \times 24$ feature map from a $336 \times 336$ input. We compare two downsampling methods: a learnable lightweight CNN (LDPv2 from (Chu et al., 2024)) and a non-learnable

| Image Pooling | | | Video Pooling | |
|---|---|---|---|---|
| CNN | $4 \times 4$ | $2 \times 2$ | Time | Space |
| 3.25 | 3.25 | **3.28** | 3.23 | **3.28** |

adaptive average pooling with a $2 \times 2$ kernel. Results indicate that adaptive pooling performs better than CNN. A $4 \times 4$ adaptive pooling was also tested but showed inferior performance.

Similarly, we ablate the pooling choice for the video encoder, which takes an input of size $T \times 224 \times 224 \times C$ and outputs a feature map of $T \times 16 \times 16 \times d$. We compare two pooling strategies: time pooling across the temporal dimension to reduce the feature map to $1 \times 16 \times 16 \times d$, and space pooling across the spatial dimension with a $2 \times 2$ kernel. Results shows that space pooling effectively preserves temporal information and yields better results.

**VCG+ 112K:** To demonstrate the effectiveness of `VCG+112K`, we train `VideoGPT+` with and without it and report its impact on the performance across

| VCG+ 112K | VCG | MVBench | VCG-Div | VideoMME |
|---|---|---|---|---|
| ✓ | 3.17 | 58.7 | 2.4 | 46.2 |
| ✗ | 3.28 | 58.8 | 2.5 | 48.7 |

multiple benchmarks, including VCGBench, MVBench, VCGBench-Diverse and VideoMME. On VCGBench, our data improves performance, particularly in detail orientation (DO) and temporal understanding (TU). The performance on MVBench shows minimal gains when incorporating the VCG+112k data. This is attributed to the distribution differences, as MVBench predominantly includes short videos averaging 5-40 seconds, whereas the VCG+112k dataset comprises videos from ActivityNet with an average duration of 3 minutes. However VCGBench-Diverse and VideoMME, do not include data from ActivityNet, ensuring a fair evaluation. The results shows improvement on both VCGBench-Diverse and VideoMME. This improvement can be attributed to our novel semi-automatic annotation pipeline and the enhanced instruction tuning data, which focuses on generating both detailed and concise instruction pairs. Refer to Fig. 3 for qualitative visualization of the data.

**Generalization across video conversation datasets:** To ensure a fair comparison with existing methods (Li et al., 2024; Liu et al., 2024c), we train our model on different combination of datasets for evaluation on MVBench and VCGBench. To further clarify the generalization capability of our model, we provide results on three benchmarks, VCGBench, MVBench and `VCGBench-Diverse`, using a single model trained on a combined dataset. The results demonstrate that our model maintains performance across all

| Training Data | MVBench | VCG | VCG-Div |
|---|---|---|---|
| Task-specific | 58.7 | 3.28 | 2.47 |
| Combined | 58.3 | 3.27 | 2.45 |

benchmarks, indicating its ability to generalize effectively across diverse video conversation datasets.

## 7 CONCLUSION

In this work, we introduce `VideoGPT+`, a novel video conversation model that leverages the complementary benefits of image and video encoders to achieve enhanced video understanding. `VideoGPT+` demonstrates better performance across multiple video benchmarks, owing to its dual-encoder design, lightweight visual adapters that map image/video features to a common space and a segment-wise sampling strategy that retains fine-grained temporal information. We also develop `VCG+112K`, a 112K video-instruction set using a resource-efficient semi-automated annotation pipeline that delivers further gains. Lastly, we propose `VCGBench-Diverse`, a diverse benchmark covering 18 video categories, to comprehensively evaluate video LMMs. Despite reported improvements, video LMMs still find challenges in precise action localization, understanding very long videos, and navigating long paths; areas where major improvements can unlock new applications.

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

SUPPLEMENTAL MATERIAL

We provide supplementary material for a deeper understanding and more analysis related to the main paper, arranged as follows:

1. Qualitative results (Appendix A)

2. Additional Implementation Details (Appendix B)

3. Additional ablations (Appendix C)

4. GPT Prompts (Appendix D

5. Ethics and societal impact (Appendix E)

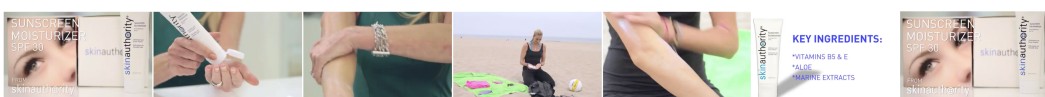

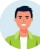 Describe the video in detail.

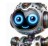 The video shows a woman applying a cream to her face and then rubbing it in. She then applies a serum to her face and rubs it in. The woman then applies a sunscreen to her face and rubs it in. The video also shows the woman applying a cream to her arm and rubbing it in.

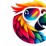 The video starts with a close-up shot of a bottle of sunscreen, followed by a woman applying it to her skin. The camera then pans out to show the woman sitting on the beach, applying the sunscreen to her arms. The sunscreen is shown again in a close-up shot, highlighting its texture and consistency. The video then shows a series of close-up shots of the sunscreen, with text appearing on the screen to highlight its key ingredients. Overall, the video showcases the importance of using sunscreen to protect the skin from harmful UV rays, with a focus on the key ingredients that make up the product.

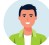 Generate a creative advertisement feature about this product that can be used to post on the product's instagram page.

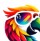 The creative advertisement feature for this product could be a video showcasing the product's benefits and how it can be used. The video could include a woman demonstrating how to use the product and the various ways it can be applied.

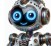 Introducing the ultimate sun protection solution for your skin! Our new sunscreen moisturizer is the perfect blend of protection and hydration for your skin. With its broad-spectrum protection, you can enjoy the sun without worrying about harmful UV rays. Plus, our moisturizing formula keeps your skin looking and feeling healthy and radiant. Say goodbye to dry, flaky skin and hello to a smooth, supple complexion. Don't forget to apply it generously and reapply every two hours for maximum protection. Your skin deserves the best, and our sunscreen moisturizer is the perfect way to give it the care it needs. Try it today and see the difference for yourself!

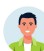 What is the SPF value of the sunscreen and what is the name of the sunscreen?

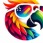 The sunscreen in the video has an SPF value of 30. The name of the sunscreen shown in the video is Key Benefits.

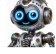 The sunscreen has an SPF value of 30 and the sunscreen is called "Skin Authority Sunscreen Moisturizer"

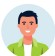 User    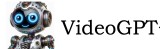 VideoGPT+    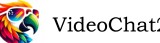 VideoChat2

Figure 5: **Qualitative comparison of VideoGPT+ with VideoChat2.** Our `VideoGPT+` demonstrates superior temporal understanding by correctly identifying multiple events in the video, effective reasoning in generating a creative advertisement, and accurate spatial understanding by identifying the SPF value and brand name of the sunscreen.

## A    QUALITATIVE RESULTS

We provide a qualitative comparison of our `VideoGPT+` with the previous state-of-the-art approach, VideoChat2 (Li et al., 2024), in Fig. 5. The example shows an advertisement video for sunscreen, where multiple scene changes are present. The video starts with a close-up view of the sunscreen, followed by a woman applying sunscreen on her hand, then applying sunscreen near a beach. The woman is then seen applying sunscreen on her arms, and finally, the video shows the key ingredients of the sunscreen and ends with the cover of the sunscreen.

As shown in Fig. 5, our `VideoGPT+` correctly identifies the events present in the video and provides a detailed and accurate description. On the other hand, VideoChat2 struggles to accurately capture all the events. Further, our model generates an advertisement post highlighting one of the unique features of the sunscreen shown in the video, namely that it functions as both sunscreen and moisturizer. Lastly, our `VideoGPT+` correctly identifies the SPF value and brand name of the sunscreen, while VideoChat2 struggles to correctly identify the brand name. We present further comparison in Fig. 6-7.

## B    ADDITIONAL IMPLEMENTATION DETAILS

In this section, we provide additional implementation details regarding our training setup and compute requirements. All of our experiments are conducted using 8xA100 40GB GPUs. The training for VCGBench experiments takes around 12 hours to complete, while the training for MVBench experiments finishes in around 10 hours. We use the model trained for the VCGBench task to evaluate on `VCGBench-Diverse` and zero-shot question-answering benchmarks. All of our training and evaluation codes, pretrained models and dataset will be publicly released.

## C    ADDITIONAL ABLATIONS

**Feature concatenation strategy:** We conduct an ablation study to determine the optimal order in which image and video features should be input to the LLM. Specifically, we perform two experiments. In the first experiment, image and video features are extracted for each video segment and concatenated in an interleaved manner before sending as input to the LLM. For example, the video is divided into segments of equal size, and then the image and video features

| Feature Concatenation | VCGBench | | | | | Avg. |
|---|---|---|---|---|---|---|
| | CI | DO | CU | TU | CO | |
| Interleaved | 3.25 | 3.17 | 3.72 | 2.78 | 3.39 | 3.26 |
| Sequential | 3.27 | 3.18 | 3.74 | 2.83 | 3.39 | 3.28 |

Table 5: **Ablation on Feature Concatenation Strategy.** Performance comparison between interleaved and sequential feature concatenation strategies. The sequential feature concatenation performs better.

from each segment are concatenated and input to the LLM. In the second experiment, we first place all the image features followed by all the video features. The results shown in Table 5, indicate that the sequential design, where the image features are placed first followed by the video features, yields better performance. This can be justified by the fact that we use different visual adapters for image and video features, so interleaving the features from both modalities can create a larger distribution shift, hindering the learning process.

**Generalization of `VideoGPT+` to other LLMs :** We train `VideoGPT+` with different LLMs including Vicuna 7B and 13B (Chiang et al., 2023) and LLaMA-3 8B (AI, 2024). We observe slight improvements in VCGBench scores when training using better LLMs, including Vicuna 13B and LLaMA-3 8B models.

| LLM | VCGBench | | | | | Avg. |
|---|---|---|---|---|---|---|
| | CI | DO | CU | TU | CO | |
| Phi3-Mini-3.8B | 3.27 | 3.18 | 3.74 | 2.83 | 3.39 | 3.28 |
| Vicuna-7B | 3.22 | 3.14 | 3.69 | 2.65 | 3.46 | 3.23 |
| Vicuna-13B | **3.30** | 3.20 | **3.75** | 2.77 | **3.48** | **3.30** |
| LLaMA3-8B | 3.29 | **3.21** | 3.73 | **2.86** | 3.38 | 3.29 |

## D  GPT PROMPTS

In this section, we provide the GPT prompts used for the following tasks: (i) Dense video description generation for `VCG+112K`, (ii) Question-answer generation for `VCG+112K` and (iii) Question-answer generation for `VCGBench-Diverse`.

**Dense Video Description Generation for VCG+ 112K:** To generate dense video captions, we provide GPT-4 with a concise ground truth caption of the video and detailed frame-level captions of the key-frames generated from LLaVA-v1.6 (Liu et al., 2024a). GPT-4 is then prompted to combine this information into a detailed caption for the entire video. As illustrated in Fig. 8, the prompt includes clear instructions to eliminate any conflicting information, ensuring an accurate and detailed caption.

**Question-answer generation for VCG+ 112K:** After generating detailed video descriptions using GPT-4, we use GPT-3.5 to create question-answer pairs for instruction tuning. Fig. 9 shows the prompt to generate detailed summary question-answer pair using the ground truth caption and the dense description of the video.

**Question-Answer Generation for  VCGBench-Diverse:** We provide prompts used to generate comprehensive question-answer pairs for `VCGBench-Diverse`. As illustrated in Fig. 10, the questions are generated in three categories: temporal, spatial, and reasoning. Similar prompts are used to generate consistency and summary questions, offering an extensive evaluation protocol for `VCGBench-Diverse`.

## E  ETHICS AND SOCIETAL IMPACT

We use multiple open-source video datasets including ActivityNet Fabian Caba Heilbron & Niebles (2015), WebWid Bain et al. (2021), MSVD-QA Xu et al. (2017), MSRVTT-QA Xu et al. (2017), TGIF-QA Jang et al. (2019), HDVILA Xue et al. (2022), MPII Andriluka et al. (2014), YouCook2 Zhou et al. (2018), UCF Crime Sultani et al. (2018), and STUD Traffic Xu et al. (2021) in our work. To the best of our knowledge, the dataset does not portray any strong biases or discrimination. We urge for the responsible use of `VideoGPT+` and `VCG+112K`, promoting research progress while safeguarding privacy.

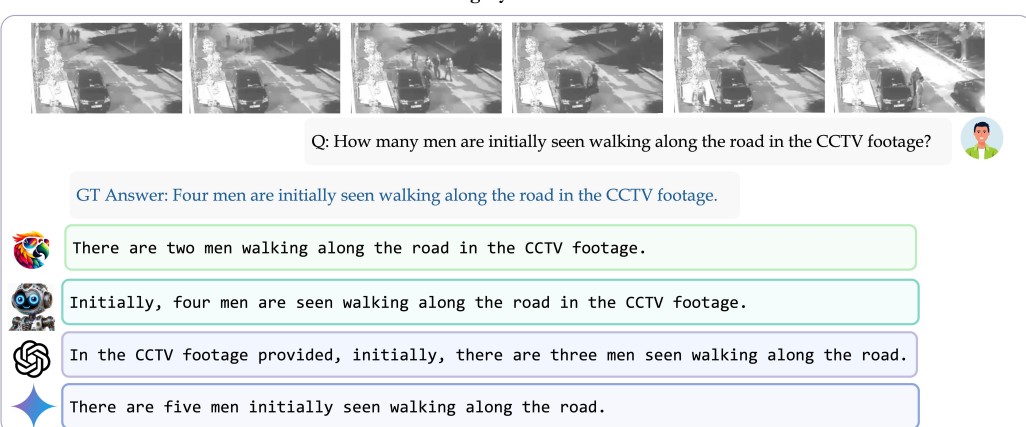

Q: Why did the car with the dashboard camera shift out of the road?

GT Answer: The car with the dashboard camera shifted out of the road because the white pick-up truck tried to change lanes, causing the car to avoid a potential collision.

The car with the dashboard camera shifted out of the road because it was being driven by a person who was not paying attention.

The car with the dashboard camera shifted out of the road to avoid a collision with another vehicle.

The car with the dashboard camera shifted out of the road due to a collision with a white pickup truck.

The car did not move out of its lane. A white pickup truck attempts to move into the lane, but then brakes to avoid a collision with a large truck in the lane, staying in its lane.

**Category: Traffic**

Q: Why does the video highlight these specific bodybuilders' cases?

GT Answer: The video highlights these specific bodybuilders' cases to underscore the potential dangers and severe health repercussions associated with the misuse of steroids and hormones within the bodybuilding community, serving as cautionary tales.

The video highlights these specific bodybuilders' cases to showcase their achievements and success in their respective fields.

The video highlights these cases to showcase the impact of substance abuse on bodybuilders' health and careers, emphasizing the consequences of such actions.

The video highlights these specific cases to discuss the extreme measures some individuals have taken for bodybuilding, which have led to significant health risks, reputational damage, and unfortunate events.

The video highlights the cases of bodybuilders who suffered adverse effects from the use of steroids.

**Category: Education**

Q: How many men are initially seen walking along the road in the CCTV footage?

GT Answer: Four men are initially seen walking along the road in the CCTV footage.

There are two men walking along the road in the CCTV footage.

Initially, four men are seen walking along the road in the CCTV footage.

In the CCTV footage provided, initially, there are three men seen walking along the road.

There are five men initially seen walking along the road.

**Category: Surveillance**

User    VideoChat2    VideoGPT+    GPT-4V    Gemini-Pro-V

Figure 6: **Qualitative comparison from `VCGBench-Diverse` of VideoGPT+.** We show qualitative comparison of `VideoGPT+` with VideoChat2 and propriety models GPT-4V and Gemini-1.5-Pro-V from three different categories including traffic, education and surveillance from `VCGBench-Diverse`.

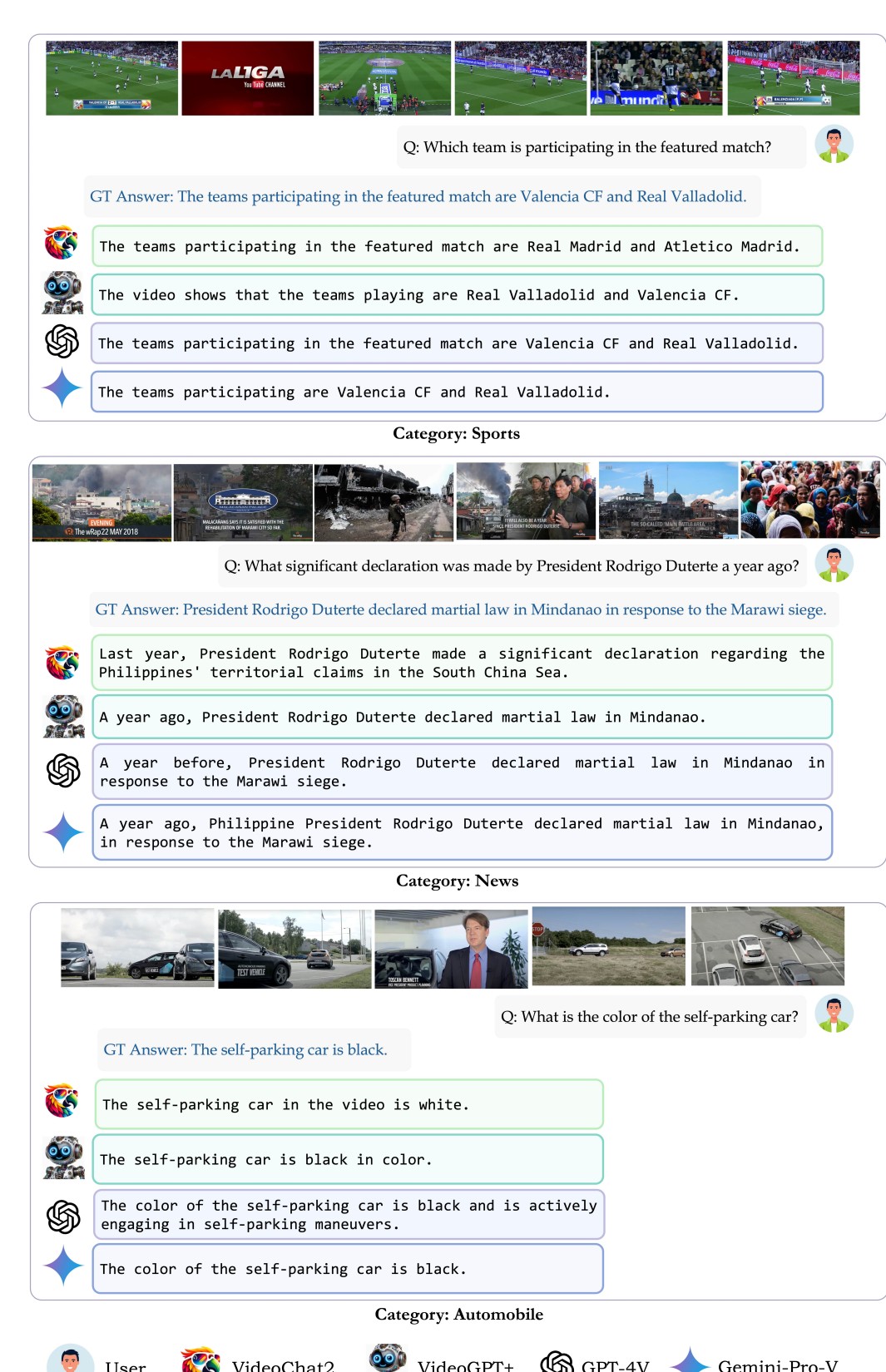

Figure 7: **Qualitative comparison from** `VCGBench-Diverse` **of VideoGPT+.** We show qualitative comparison of `VideoGPT+` with VideoChat2 and propriety models GPT-4V and Gemini-1.5-Pro-V from three different categories including sports, news and automobiles videos from `VCGBench-Diverse`.

```
Generate a detailed and accurate description of a video based on the given ground-truth video
caption and multiple frame-level captions. Use the following details to create a clear and complete
narrative:

Ground-truth Video Caption: [Ground-truth caption here]

Frame-level Captions: [Frame-level caption 1]; [Frame-level caption 2]; [Frame-level caption 3]; ..

Instructions for writing the detailed description:
    1. Focus on describing key visual details such as appearance, motion, sequence of actions,
    objects involved, and interactions between elements in the video.
    2. Check for consistency between the ground-truth caption and frame-level captions, and
    prioritize details that match the ground-truth caption. Ignore any conflicting or irrelevant
    details from the frame-level captions.
    3. Leave out any descriptions about the atmosphere, mood, style, aesthetics, proficiency, or
    emotional tone of the video.
    4. Make sure the description is no more than 20 sentences.
    5. Combine and organize information from all captions into one clear and detailed description,
    removing any repeated or conflicting details.
    6. Emphasize important points like the order of events, appearance and actions of people or
    objects, and any significant changes or movements.
    7. Do not mention that the information comes from ground-truth captions or frame-level captions.
    8. Give a brief yet thorough description, highlighting the key visual and temporal details while
    keeping it clear and easy to understand. Use your intelligence to combine and refine the
    captions into a brief yet informative description of the entire video.
```

Figure 8: **Prompt for Dense Video Captions Generation for VCG+ 112K.** We use GPT-4 to generate detailed video captions using concise ground truth and frame-level detailed captions.

```
# System Prompt
You are an AI assistant tasked with generating questions and answers about video content to create a
video instruction tuning dataset. Your goal is to extract detailed visual and temporal information
from the video, ensuring the explanations are comprehensive enough for someone to understand the
entire sequence of events in the video.

##TASK:
1. Users provide a video ground truth caption and a detailed description.
2. Generate three questions that effectively prompt a detailed description of the entire video
content and sequence of events.

------
##INSTRUCTIONS:
- Ensure each question targets the goal of generating a detailed description of the entire video
from start to end.
- Avoid questions that focus on small parts, less relevant details, or abstract concepts such as
logical reasoning, attention to subtle details, overall aesthetic.
- Every answer must include all the details from the ground truth caption and integrate additional
specifics from the detailed description.
- Focus on visual and temporal details.

##SAMPLE QUESTIONS:
- Can you describe the entire video in detail from start to finish?
- What happens throughout the entire video, including all key actions and events?
- Could you provide a detailed walkthrough of the entire video?

# User Prompt:
The video ground truth caption is: [Ground-truth caption here]. The noisy detailed description is:
[Dense description here].

Generate three questions and answers about the entire content and sequence of events in the video.
Each question should aim to elicit a comprehensive description of the full sequence of events in the
video from start to finish. Each answer must include all the details from the ground truth caption
and integrate additional specifics from the detailed description. Format the output as a list of
dictionaries in JSON style, with each dictionary containing a 'Q' key for
the question and an 'A' key for the answer. For example:

[{'Q': 'Your first question here...', 'A': 'Your first answer here...'}, {'Q': 'Your second
question here...', 'A': 'Your second answer here...'}, {'Q': 'Your third question here...', 'A':
'Your third answer here...'}].

Most importantly, every answer must provide a full understanding of the video by incorporating ALL
the details from the ground truth caption and additional specifics from the detailed description.
```

Figure 9: **Prompt for Question-answer generation for VCG+ 112K**. We use GPT-3.5 to generate question-answer pairs for instruction tuning using the concise video ground truths and detailed video descriptions.

```
# System Prompt:
You are an AI assistant tasked with generating questions and detailed answers based on a video
description. Your goal is to extract important information from the video content, focusing on
temporal events, visual details, and reasoning behind actions.

##TASK:
You will receive a video description, and based on it, you must generate a set of questions and
answers in three distinct categories:
1. Temporal - These questions should focus on the sequence and timing of events. Use approximate
time references where necessary.
2. Spatial - These questions should address visual aspects such as appearance, objects, colors,
attire, displayed texts, number of objects or people, location, and other significant visual
details.
3. Reasoning - These questions should delve into the actions, motivations, and consequences as
depicted in the video description.

##INSTRUCTIONS:
- Each question must directly relate to and be answerable by the provided video description. Avoid
assumptions and fabrication of details not present in the description.
- Provide clear, unambiguous questions that allow for definitive answers based on the description.
- If the video description does not contain enough information to formulate a question in any
category, do not include a question for that category.

##SAMPLE QUESTIONS:
- Temporal: Describe the entire process the person goes through from start to finish or What happens
at the beginning of the video? or What does the person do right after the dog appears?
- Spatial: Can you provide a detailed description of the appearance and activities of all
individuals or What is the color of the main character's shirt? or What is the name of the drink on
the bottle? How many people are at the table?
-  Reasoning: What action does the coach take after the whistle blows? or Why did the player throw
   the ball? or Who is John Davis in the video?

# User Prompt:
The video description is: [Dense description here].

Format the output as a dictionary in JSON style, with each key representing a question category and
containing a sub-dictionary with 'Q' for the question and 'A' for the answer. Example output with
all three categories filled:

{'temporal': {'Q': 'Temporal question here...', 'A': 'Answer here...'},
  'spatial': {'Q': 'Spatial question here...', 'A': 'Answer here...'},
  'reasoning': {'Q': 'Reasoning question here...', 'A': 'Answer here...'}}.

If a category cannot be filled:

{'temporal': {'Q': 'Describe the sequence of events in the video.', 'A': 'The video starts
with...'},'spatial': {'Q': 'What is the main character wearing?', 'A': 'The main character is
dressed in...'}} # reasoning omitted due to lack of information

Importantly, the answers MUST extract information DIRECTLY from the given description. Do not
include categories that cannot be filled based on the video description alone.
```

Figure 10: **Prompt for Question-Answer Generation for VCGBench-Diverse.** We use GPT-3.5 to generate temporal, spatial, and reasoning question-answer pairs.

