# OpenReview forum: "VideoGPT+: Integrating Image and Video Encoders for Enhanced Video Understanding"
_ICLR.cc/2025/Conference — ICLR 2025 Conference Withdrawn Submission_

### Official Review · Reviewer_bHZV · 2024-10-31

**Soundness:** 2
**Presentation:** 3
**Contribution:** 2
**Rating:** 3
**Confidence:** 5

**Summary:**

This paper studies the problem of video question-answering (video QA).

The authors claim three novelties/contributions:
1. A video-language model that integrates frame-level image features, clip-level video features and an LLM (VideoGPT+)
2. A ‘novel semi-automatic annotation pipeline’ to generate instruction-tuning data (VCG+ 112K)
3. A new evaluation benchmark for video QA (VCGBench-Diverse)

**Strengths:**

The paper studies an interesting problem, video question-answering. This is a very active topic in the community that’s advancing rapidly. The potential applications of this technology, when successful, will be interesting.

The paper is generally easy to follow, but I do have some comments on some presentation choices, as outlined later in this review.

The first half of the Experiments section shows strong empirical performance on various benchmarks.

**Weaknesses:**

My first key concern is the lack of acknowledgement from the authors that the idea of combining video features and image features has been studied in the vision community for a very long time. There is absolutely no discussion on any prior work that has taken this approach, which was very surprising to me. This may give the impression that the authors are first to combine these two feature types, but this is absolutely not the case in the video understanding literature. On the other hand, this paper may indeed be the first in which a video+LLM method combines image and video features, but given the vast amount of prior work that have studied such combinations without LLMs, it makes this contribution not very surprising/significant.

Here are some examples of methods that combine image and video features for video understanding. I’m sure the list could be made much better with some more time.
* [Two-Stream Convolutional Networks for Action Recognition in Videos. Simonyan & Zisserman. 2014]
* [SlowFast Networks for Video Recognition. Feichtenhofer, Fan, Malik & He. 2019]
* [Camera Motion and Surrounding Scene Appearance as Context for Action Recognition. Heilbron, Thabet, Niebles & Ghanem. 2014]
* [A Closer Look at Spatiotemporal Convolutions for Action Recognition. Tran, Wang, Torresani, Ray, LeCun & Paluri. 2017]
* [Rethinking Spatiotemporal Feature Learning: Speed-Accuracy Trade-offs in Video Classification. Xie, Sun, Huang, Tu & Murphy. 2018]
* [Action Recognition and Detection by Combining Motion and Appearance Features. Wang, Qiao, Tang. 2014]
* [Joint Action Recognition and Pose Estimation From Video. Nie, Xiong &  Zhu. 2015.]



My second key concern with this paper is its lack of focus. The three contributions are somewhat independent, so in a sense it almost feels like they belong to three different papers. The story of image+video features for the model is disconnected from the story of the instruction tuning data, and these two stories are disconnected from the story of the new evaluation benchmark. There is no cohesiveness, so it is unclear what the key research question and hypothesis is here.

In the experiments, there are ablations for the image+video feature combination (463-469). This study includes the new instruction tuning data, so it is easy to compare performance with other work: is the difference in performance due to the feature integration or because this model has additional training data? Here, contributions 1 (model) and 2 (instruction tuning data) are entangled.

There is also an ablation on the instruction tuning data (lines 506-518). Here the comparison is based on the full model that integrates video+image features. Again, is the model without the data competitive? This is not clearly presented here, which makes these two contributions entangled.

While the experiments entangle contributions 1 and 2, their design and implementation are actually disconnected. Or at least, there’s no clear presentation in the paper of how model design and instruction data design influence each other, which makes the reader think they are actually unrelated.

A similar problem happens with the evaluation benchmark. The paper is lacking in discussing how this benchmark relates to the rest of the paper. To be clear, there are cases in which a paper sets up a new problem, and thus needs to introduce a benchmark to evaluate the problem, as well as a method to tackle the problem. Here, the task is well studied (video QA), the key idea of the model innovation is about image+video features, and the benchmark seems to be about domain diversity. In that sense, it is unclear to me why this benchmark and model belong together in the same manuscript.


Experimental support and validation of claims:
1. Claim 1: image+video feature fusion. The authors design a way to fuse image and video features which seems reasonable to me. There’s no discussion on any other prior work that fuses video and image features, and no comparison to alternative design choices for this fusion. The only supporting evidence is an ablation on using image-only, video-only and dual features (lines 463-496) without comparison to alternative designs.
2. Claim 2: ‘novel semi-automatic annotation pipeline’ (line 088) to construct an instruction tuning dataset. There is no discussion in the paper of what makes this data pipeline novel. I understand it’s new data, but the claim here is that the pipeline is novel. If the claim is instead about the generated data instead, then there should be evaluations on the data quality, but these are not provided either. Section 4 seems to emphasize the ‘improvement’, ‘high-quality’ of this data, so not having a direct quality evaluation makes this claim weak. I acknowledge there is an ablation on removing this set from the instruction tuning phase (lines 506-518), but since training data size is not controlled, the performance differences could be attributed to the change in training data size instead of the actual quality of the data.
3. Claim 3: New benchmark. This is useful for the community, but there is very little information on what makes this evaluation dataset new/novel. I acknowledge the discussion on 18 distinct domains (line 301-319), but there’s no quantitative comparison on what and how many domains are covered on previous datasets. Since the claim here appears to be about increased diversity, there should be some metric and comparison to support it.

Discussion of Related Work:

Missing from the related work section is discussion about: (a) methods for fusion of multiple features, (b) methods that combine image+video features for video understanding, (c) methods for semi-automated dataset annotation.

Another issue in the related work section is at times (for instance lines 120-125) simply a list of prior work without clear insights of what’s their limitation, or how they relate to this paper. Creating a relationship to prior work is important for the reader to better appreciate the work.

Presentation:
1. Paragraph 1 from the introduction (lines 037-048) makes assertions that are written generally about “video understanding”, but instead apply only to video-language models. To correct this, I suggest modifying the first sentence to include the scope more clearly on ‘video-language’, rather than general video understanding.
2. Through the paper, the authors talk about ‘local features’ and ‘global features’ (line 051, Figure 2, line 205, etc). My opinion is that this is a misnomer. What the authors name as ‘local features’ are really static frame-level features, and ‘global features’ are really dynamic clip-level features. That is, these dynamic clip-level features are not really global for the reasons claimed by the authors: 1) the frame resolution is not too different from the static image features (336x336 vs 224x224, line 346), and these features also come in a spatial grid so they are not spatially global either. I suggest removing the local/global name and instead use static/dynamic or image/clip features.

**Questions:**

Other suggestions for improvement:
- Additional ablations that would be helpful: use one of the base models mentioned in the paper (e.g. video-ChatGPT) without modification and augment its training with your new instruction tuning data. This would help showing the benefits of the new instruction tuning data.
- Line 053: “uniform sampling” -> uniform frame sampling
- Line 160: ”uniform sampling” -> dense uniform frame sampling
- Line 184: “trained with sparse frames” -> do you mean they are trained with a small number of frames? Or with a small frame rate? This could be made more clear.
- Split experiments in two subsections: first subsection would contain performance evaluation of the full pipeline (model+data) on various existing benchmarks, plus the new benchmark proposed here (lines 340-462). The second subsection would contain all ablations (lines 463-526).
- Section 4: This section does not clearly state what is the source of videos for VCG+ 112K.

---

### Official Review · Reviewer_yRXZ · 2024-11-02

**Soundness:** 3
**Presentation:** 3
**Contribution:** 3
**Rating:** 5
**Confidence:** 5

**Summary:**

The paper introduces VideoGPT+, a novel video understanding model that integrates image and video encoders to enhance video comprehension by leveraging detailed spatial information and temporal context. It presents VCG+ 112K, a new dataset developed through a semi-automatic annotation pipeline, which improves model performance. Additionally, the paper proposes VCGBench-Diverse, a benchmark covering 18 video categories, for a comprehensive evaluation of video LMMs.

**Strengths:**

1. The paper's strengths include its innovative dual-encoder design that effectively combines image and video encoders for richer spatiotemporal understanding of videos.
2. The introduction of the VCG+ 112K dataset, developed through a semi-automatic annotation pipeline, enhances the model's performance by providing dense video captions and reasoning-based QA pairs.
3. Furthermore, the proposal of the VCGBench-Diverse benchmark allows for a more comprehensive evaluation of video LMMs across diverse video types and dynamics.

**Weaknesses:**

The paper has the following weaknesses.
1. The simultaneous use of both an image and a video encoder increases the number of parameters in the model, making it difficult to determine whether the effectiveness of the paper is accurate or if the performance improvement is simply due to the increased parameters. Are there any studies comparing the effects of using only an image encoder versus using both an image and a video encoder?

2. The image encoder and the video encoder are independent of each other, with no interaction between them. Can you compare the performance effects between having information interaction and not having information interaction between them?

3. The paper lacks some theoretical derivations.

**Questions:**

1. Are there any studies comparing the effects of using only an image encoder versus using both an image and a video encoder?

2. Can you compare the performance effects between having information interaction and not having information interaction between them?

---

### Official Review · Reviewer_yT2F · 2024-11-02

**Soundness:** 3
**Presentation:** 2
**Contribution:** 2
**Rating:** 3
**Confidence:** 5

**Summary:**

The authors of the paper propose a new video large language model, VideoGPT+, by (1) using two encoders—an image encoder and a video encoder—to extract visual features, and (2) introducing a new video instruction tuning dataset called VCG+112K, which contains 112K samples generated using a semi-automatic annotation pipeline based on PySceneDetect for keyframe extraction, LLaVA-v1.6 for frame description generation, GPT-4 for detailed video description generation, and finally GPT-3.5 for QA pair generation.

Additionally, they introduce a new benchmark called VCGBench-Diverse, consisting of 4,354 question-answer pairs for 877 videos sourced from HDVILA, MPII, YouCook2, UCF Crime, and STUD Traffic. A human annotation process assisted by GPT-3.5 is used to obtain the QA pairs for this benchmark.

**Strengths:**

1. The model VideoGPT+ is a 3.8B-scale model. The VideoGPT+ model along with the VCG+112K dataset will be valuable resources once refined and released.

2. The primary design choice in this work—using both an image encoder and a video encoder to develop a video large language model—is reasonable and straightforward.

**Weaknesses:**

1. **The novelty and technical depth of this paper are limited. While the paper follows empirical best practices for constructing a multimodal large language model, it lacks the original and novel ideas expected for a conference like ICLR.** The main architectural change introduced in VideoGPT+ is the use of a dual-encoding scheme; however, this approach is neither particularly innovative nor original. In the multimodal large language model domain, the use of two or multiple encoders to extract visual features has been widely explored [1-5]. In terms of ideas, I don’t see a fundamental difference, notable innovation, or new insights.

2. **(a) The introduced benchmark, VCGBench-Diverse, is limited: it is small (containing only 877 videos) and supports only open-ended question answering.** The authors claim that a multiple-choice question-answering format introduces bias and fails to capture the model's true understanding (Lines 136-138), which I disagree with, as open-ended QA, relying on LLM-as-a-judge, often leads to more problematic evaluations compared to multiple-choice QA. In multimodal large language model evaluation, the trend favors using multiple-choice QA over open-ended QA for robust and reliable benchmarking, or incorporating multiple task formats (e.g., TempCompass).
**(b) There are many benchmarks for video conversation models nowadays. It is also unclear how VCGBench-Diverse is unique compared to existing benchmarks**, including those mentioned in this paper, such as VCGBench, MVBench, and VideoMME, as well as others not mentioned, such as TempCompass [6], CVRR-ES [7], Vinoground [8], VideoVista [9], and more [10-14].

3. **State-of-the-art (SOTA) works**, such as LLaVA-NeXT-Video [15], LLaVA-OneVision [16], PLLaVA [17], SlowFast-LLaVA [18], Tarsier [19], MiniGPT4-Video [20], ShareGPT4Video [21], VideoLLaMA 2 [22], etc., **are missing**. The SOTA model comparisons are unconvincing due to the absence of these SOTA models and different training data recipes are used across models. Moreover, the **ablation studies are performed on a selectively chosen and inconsistent subset of evaluation benchmarks**.

4. **Latency is not discussed.**

5. **Clarity of the paper could be improved.**


References:

[1] BRAVE: Broadening the Visual Encoding of Vision-Language Models

[2] Eagle: Exploring The Design Space for Multimodal LLMs with Mixture of Encoders

[3] SPHINX: The Joint Mixing of Weights, Tasks, and Visual Embeddings for Multi-modal Large Language Models

[4] Eyes Wide Shut? Exploring the Visual Shortcomings of Multimodal LLMs

[5] Grounded-VideoLLM: Sharpening Fine-grained Temporal Grounding in Video Large Language Models

[6] TempCompass: Do Video LLMs Really Understand Videos?

[7] How Good is my Video LMM? Complex Video Reasoning and Robustness Evaluation Suite for Video-LMMs

[8] Vinoground: Scrutinizing LMMs over Dense Temporal Reasoning with Short Videos

[9] VideoVista: A Versatile Benchmark for Video Understanding and Reasoning

[10] AutoEval-Video: An Automatic Benchmark for Assessing Large Vision Language Models in Open-Ended Video Question Answering

[11] VideoBench: https://github.com/PKU-YuanGroup/Video-Bench

[12] MLVU: A Comprehensive Benchmark for Multi-Task Long Video Understanding

[13] LongVideoBench: A Benchmark for Long-context Interleaved Video-Language Understanding

[14] TemporalBench: Benchmarking Fine-grained Temporal Understanding for Multimodal Video Models

[15] LLaVA-NeXT: Tackling Multi-Image, Video, and 3D in Large Multimodal Models

[16] LLaVA-OneVision: Easy Visual Task Transfer

[17] PLLaVA: Parameter-free LLaVA Extension from Images to Videos for Video Dense Captioning

[18] SlowFast-LLaVA- A Strong Training-Free Baseline for Video Large Language Models

[19] Tarsier: Recipes for Training and Evaluating Large Video Description Models

[20] MiniGPT4-Video: Advancing Multimodal LLMs for Video Understanding with Interleaved Visual-Textual Tokens

[21] ShareGPT4Video- Improving Video Understanding and Generation with Better Captions

[22] VideoLLaMA 2 Advancing Spatial-Temporal Modeling and Audio Understanding in Video-LLMs

**Questions:**

1. Since your main modeling contribution is the use of dual encoders, for fair comparisons, could you use the data recipe and model design from an existing work (e.g., VideoLLaVA) and apply your dual encoders to demonstrate the effectiveness of your dual-encoder design across evaluation benchmarks? Similarly, could you train another model architecture on VCG+112K to demonstrate the effectiveness of VCG+112K? That would make the paper's claims more reliable.

2. What is the video source for VCG+112K? Is it ActivityNet? Are the videos in VCG+112K the same set of videos as those in VideoInstruction100K? How many videos are included in VCG+112K?

3. Lines 267–269 mention that the detailed video descriptions generated by GPT-4 include a timeline of events, actions, object attributes, and scene settings. Could you qualitatively show some examples of video descriptions obtained after this step? Since a timeline of events is included, does this imply that a subset of the resulting VCG+112K data could be used to train models like Grounded-VideoLLM for tasks such as temporal grounding? Would it be advisable to choose the dense captions from VCG+112K over other existing video description/caption datasets for training a video description/caption model?

4. Lines 473–475 mention that the video-only model performs better than the image-only model on MVBench action categories. How does the dual-encoder design perform in comparison?

5. For the ablation 'without segment-wise sampling,' did you use uniform frame sampling instead? For the ablation 'without adaptive token pooling,' it is mentioned that this restricts the model to fewer frames. How many frames were used for this ablation? More implementation details are needed for the ablations.

6. In the table showing the results of the VCG+112K ablation study, does the second row indicate the use of VCG+112K in training? The table design is problematic because the second row shows better performance but is marked with a cross mark, suggesting that VCG+112K was not used.

7. Line 524 mentions a combined dataset. Does this mean that you combined all the training data listed in lines 353 to 364? For datasets with multiple splits mentioned in lines 353 to 364, did you use only the training split? Did you use the full dataset for each one mentioned in lines 353 to 364, or was any sampling performed when designing your training data recipe?

8. How is VCGBench-Diverse unique compared to existing benchmarks?

9. How do the new design choices affect the training and inference costs? How does the training data affect the training time?

Minor comments:

In Figure 1 caption: "VideoGPT+ permors better" -> "VideoGPT+ performs better"

---

### Official Review · Reviewer_aArr · 2024-11-03

**Soundness:** 2
**Presentation:** 3
**Contribution:** 2
**Rating:** 3
**Confidence:** 4

**Summary:**

The paper presents VideoGPT+, a multimodal large language model (LMM) designed to enhance video understanding by integrating image and video encoders. VideoGPT+ combines an image encoder for spatial details and a video encoder for temporal context. Using a segment-wise sampling approach and adaptive pooling, the model improves on multiple benchmarks, demonstrating significant gains in video captioning, temporal understanding, and question-answering tasks. Additionally, the authors introduce VCG+112K, a semi-automatically annotated dataset, and VCGBench-Diverse, a comprehensive benchmark spanning 18 video categories, to further assess model generalization in video conversation tasks​.

**Strengths:**

1. The paper is well-written, providing clear explanations of both the model architecture and its rationale.

2. The introduction of VCGBench-Diverse adds a valuable resource for evaluating video models across diverse domains.

3. The model demonstrates improved performance on key benchmarks, indicating robust capabilities in video understanding tasks.

**Weaknesses:**

1. The approach might be considered trivial, primarily relying on combining two powerful existing encoders without introducing new theoretical insights to advance the field.

2. The paper lacks comparisons with significant baselines, specifically LongVA [1], InternVL [2] and Kangaroo [3], which could provide a clearer context for evaluating its contributions​.


[1] Zhang, Peiyuan, et al. "Long Context Transfer from Language to Vision." arXiv preprint arXiv:2406.16852 (2024).


[2] Chen, Zhe, et al. "InternVL: Scaling up Vision Foundation Models and Aligning for Generic Visual-Linguistic Tasks." arXiv preprint arXiv:2312.14238 (2023).

[3] Mingo Hoffman, Enrico, et al. "Modeling and Numerical Analysis of Kangaroo Lower Body based on Constrained Dynamics of Hybrid Serial-Parallel Floating-Base Systems." arXiv preprint arXiv:2312.04161 (2023).

**Questions:**

1. What is the sampling frame rate for the image encoder and the video encoder? How many frames are input to each encoder in total?

2. Have the authors verified if the performance improvement is due to the increased number of vision tokens? What is the performance difference between a pure image encoder and a hybrid encoder with an equivalent number of tokens?

---

### Official Review · Reviewer_8Pn4 · 2024-11-04

**Soundness:** 2
**Presentation:** 1
**Contribution:** 2
**Rating:** 3
**Confidence:** 4

**Summary:**

The paper claims that the image encoder could capture rich spatial details while the video encoder could provide temporal contexts with sparse frames as input. Based on such observation, the authors propose to integrate both image encoder embeddings and video encoder embeddings to take both the benefits for video-language understanding. The performance on VCGBench, VCGBench-Diverse, MVBench and VideoMME shows the superiority of the trained model. The paper also proposes a new benchmark, called VCG-Diverse, posing its efforts towards question-answering of diversified video topics.

**Strengths:**

1. The idea of combining a high-resolution image encoder and a low-resolution sparse video encoder makes sense. It can somehow reduce the model training cost and inference latency, compared to full high-resolution dense video input.
2. The paper introduces new training data curated by GPT-3.5, which might serve as new training data to feed the community.
3. The paper introduces a new Video-QA benchmark using GPT-3.5 called VCGBench-Divese, which is based on human-curated descriptions.
4. Experiments on various Video-QA benchmarks show the effectiveness of the trained model.

**Weaknesses:**

1. Related recent literature is missing. The paper has two efforts, one is towards better video understanding with LLM, e.g., [1][2][3], the other one is towards video-language benchmark and training data curation [4][5]. The paper lacks good literature reviews in both directions.
2. Training data curation - necessity and fairness: To illustrate the effectiveness of the method and design. I think the experiments should be at least conducted on same data source and annotation approach. Curating new data  could be a good community contribution, however, the training data from existing works should be used to experiment to show the effectiveness of arch design.
3. Inference data curation - video sources: The usage of HDVILA is doubtable. HDVILA was introduced as a pretraining dataset and an important source of training data. Using it for general Video-QA inference purposes may lead to leakage.
4. Inference data curation - QA quality control: The VCGBench-Diverse QA is generated by GPT. How to control the quality? Is there any manual check involved?
4. Experiments - Fairness: Missing some recent works. Please include the works for fair comparison.
5. Experiments - Ablation: Adequate ablation experiments should be performed on the effectiveness of image encoder branch and video encoder branch on common video question answering benchmarks. In such way, the necessity of the dual branches can be validated.
6. Format - the tables on Pages 9 and 10 don't have captions (excluding Table 4).
7. The authors try to fit too much content into a single paper, which make the contribution of this paper ambiguous. For example, the integration of the image encoder and video encoder doesn't seem to be correlated to the data annotation process and new benchmark. I treasure their efforts, however, this might not be a good practice for me.

[1] Llava-next-interleave: Tackling multi-image, video, and 3d in large multimodal models
[2] Pllava: Parameter-free llava extension from images to videos for video dense captioning
[3] Slowfast-llava: A strong training-free baseline for video large language models
[4] ShareGPT4Video: Improving Video Understanding and Generation with Better Captions
[5] Shot2Story: A New Benchmark for Comprehensive Understanding of Multi-shot Videos

**Questions:**

1. What is the instructions provided to the annotators to obtain good captions?
2. Why not directly use the training data from plenty of existing works and see the effectiveness of the idea of integrating image encoder and video encoder?
3. In terms of zero-shot question-answering, is Activitynet data excluded from the training set?
4. Which results does Line 470-475 refer to? I don't see a cross-reference in the paper for this.
5. Are results in tables 463 and 490 comparable? I understand from Lines 480-482 that these numbers might be based on different samples. Please correct me if I'm taking it wrong.
6. By designing the new benchmarks, what are the insights and new conclusions for model design?

---

### Note · Authors · 2024-11-25

I have read and agree with the venue's withdrawal policy on behalf of myself and my co-authors.